# Immunoadsorption and Plasma Exchange in Seropositive and Seronegative Immune-Mediated Neuropathies

**DOI:** 10.3390/jcm9072025

**Published:** 2020-06-27

**Authors:** Alexander J. Davies, Janev Fehmi, Makbule Senel, Hayrettin Tumani, Johannes Dorst, Simon Rinaldi

**Affiliations:** 1Nuffield Department of Clinical Neurosciences, University of Oxford, Oxford OX3 9DU, UK; alexander.davies@ndcn.ox.ac.uk (A.J.D.); janev.fehmi@sjc.ox.ac.uk (J.F.); 2Department of Neurology, University of Ulm, 89081 Ulm, Germany; makbule.senel@uni-ulm.de (M.S.); hayrettin.tumani@uni-ulm.de (H.T.); johannes.dorst@uni-ulm.de (J.D.)

**Keywords:** Inflammatory neuropathy, chronic inflammatory demyelinating polyneuropathy, Guillain-Barré syndrome, multiple sclerosis, paranodal antibodies, plasmapheresis, plasma exchange, immunoadsorption

## Abstract

The inflammatory neuropathies are disabling conditions with diverse immunological mechanisms. In some, a pathogenic role for immunoglobulin G (IgG)-class autoantibodies is increasingly appreciated, and immunoadsorption (IA) may therefore be a useful therapeutic option. We reviewed the use of and response to IA or plasma exchange (PLEx) in a cohort of 41 patients with nodal/paranodal antibodies identified from a total of 573 individuals with suspected inflammatory neuropathies during the course of routine diagnostic testing (PNAb cohort). 20 patients had been treated with PLEx and 4 with IA. Following a global but subjective evaluation by their treating clinicians, none of these patients were judged to have had a good response to either of these treatment modalities. Sequential serology of one PNAb+ case suggests prolonged suppression of antibody levels with frequent apheresis cycles or adjuvant therapies, may be required for effective treatment. We further retrospectively evaluated the serological status of 40 patients with either Guillain-Barré syndrome (GBS) or chronic inflammatory demyelinating polyneuropathy (CIDP), and a control group of 20 patients with clinically-isolated syndrome/multiple sclerosis (CIS/MS), who had all been treated with IgG-depleting IA (IA cohort). 32 of these patients (8/20 with CIDP, 13/20 with GBS, 11/20 with MS) were judged responsive to apheresis despite none of the serum samples from this cohort testing positive for IgG antibodies against glycolipids or nodal/paranodal cell-adhesion molecules. Although negative on antigen specific assays, three patients’ pre-treatment sera and eluates were reactive against different components of myelinating co-cultures. In summary, preliminary evidence suggests that GBS/CIDP patients without detectable IgG antibodies on routine diagnostic tests may nevertheless benefit from IA, and that an unbiased screening approach using myelinating co-cultures may assist in the detection of further autoantibodies which remain to be identified in such patients.

## 1. Introduction

The inflammatory neuropathies are a heterogeneous group of disorders in which peripheral nerve function and structure are disturbed by largely ill-defined immunological mechanisms [1]. They can broadly be divided into acute and chronic forms, typified by the umbrella terms Guillain-Barré syndrome (GBS) and chronic inflammatory demyelinating polyneuropathy (CIDP), respectively. Humoral and cellular immunity are likely to play a role in the pathogenesis of both syndromes. For some clinically defined subtypes, a role for the humoral immune system and pathogenic autoantibodies appears to be more prominent [2,3], but particularly at the level of the individual patient, a direct and consistent link between the clinical syndrome, serological profile, and underlying immunopathological mechanism remains difficult to establish.

Randomised controlled trials have demonstrated that therapeutic plasma exchange (PLEx) speeds up recovery from GBS [4], and provides at least a short-term improvement in disability in CIDP [5]. In both conditions there is evidence that intravenous immunoglobulin (IVIg) has similar efficacy [6,7]. Two small, randomised studies have compared immunoadsorption (IA) with PLEx or IVIg in CIDP. Response rates to IA (6/9 using tryptophan-based columns [8] and 4/5 using protein A [9]) were not significantly different to their respective comparators. The trial comparing IA (using protein A) with IVIg had a high drop-out rate and was excluded from the relevant Cochrane review due to a high risk of bias [9]. Two further reports described the crossover from PLEx to IA in CIDP, in a single patient each, reaching opposite conclusions about which was more efficacious [10,11]. A number of retrospective case series and case reports have favourably evaluated immunoadsorption in both GBS and CIDP [12,13,14,15,16,17,18,19,20,21]. A retrospective Japanese report of IA in GBS found that patients who received IA within 6 days of onset of their neuropathy had a more rapid improvement in disability compared to those who received supportive care alone, whereas patients who received IA later than this in their disease course did not [22]. However, high-quality evidence demonstrating the efficacy of IA in the inflammatory neuropathies is lacking [23]. There is also some evidence that apheresis can improve recovery from multiple sclerosis relapses, and these approaches are often used after inadequate responses to corticosteroids [24,25].

Certain subtypes of GBS are associated with immunoglobulin (Ig) G ganglioside antibodies [26], with a handful of small studies showing an effective reduction of antibody titres using IA [19,27]. More recently a subset of CIDP-like neuropathies have been linked to predominantly IgG4-subclass antibodies directed against nodal or paranodal cell-adhesion molecules [28,29,30,31,32]. It has been speculated that patients with such antibodies may respond particularly well to selective IgG immunoadsorption [33]. A recent case series of four patients with CIDP and neurofascin-155 (NF155) antibodies reported that PLEx was effective in 3, and partially effective in 1, whilst tryptophan-based IA was ineffective in one such patient [34].

There are of course substantial differences between PLEx and IA. The former removes a broad range of circulating molecules and requires the use of replacement fluid, typically fresh frozen plasma, or albumin. Replacement fluid is not required in IA, and the range of circulating factors removed is more limited. This is advantageous in reducing complications, such as those due to the unwanted removal of coagulation factors [35], but may also lead to a loss of therapeutic effect if this depends on the removal of pro-inflammatory cytokines, or other pathogenically-relevant molecules, rather than immunoglobulins. It is also important to appreciate that there are variations in the biological effects between the different types of IA, which may also influence their clinical efficacy. For example, Yuki and colleagues have previously demonstrated that tryptophan-based columns are more effective than phenylalanine for adsorbing anti-ganglioside antibodies [36]. IA using protein A or synthetic ligands has been proposed as a method to remove a larger fraction of circulating IgG more selectively and quickly, whilst more modestly affecting IgM and IgA levels, and leaving complement, albumin and fibrinogen largely unaffected [37].

Intuitively, it may be assumed that patients who respond to “Ig-selective” IA do so because pathogenic Ig is being removed from the circulation. However, previous assessments of IA efficacy rarely report serological status. It is therefore currently unclear as to whether the presence of known serum autoantibodies in GBS and CIDP prospectively identifies a subpopulation of patients who are likely to respond more favourably to IA. It is also unclear as to whether any particular IA system or treatment programme is more likely to produce a positive outcome.

In this study we provide a retrospective evaluation of apheresis in two serologically-defined patient cohorts. We first reviewed the subjective clinician-reported overall impression of response to IA or PLEx in a cohort of neuropathy patients identified during routine diagnostic testing (PNAb cohort), and compared patients in which nodal/paranodal antibodies were or were not detected. We present the detailed case history and parallel serological analysis of a patient with NF155 antibodies who was treated with IA. Finally, we perform a retrospective analysis of the serological status of a sample of 60 patients who had been treated with IgG-depleting IA (IA cohort) and compare this with clinician-reported outcomes.

## 2. Experimental Section

### 2.1. Paranodal Antibody (PNAb) Patient Cohort

Since 2015, 88 patients with confirmed or suspected inflammatory neuropathies presenting to the neuropathy clinic in Oxford have been recruited to an observational study. This study was approved by the National Health Service (NHS) National Research Ethics Service Committee (South Central–Oxford A, 14/SC/0280). Patients recruited prior to 2017 were tested retrospectively, and those recruited from 2017 prospectively, for nodal/paranodal antibodies by the methods described in Appendix B. Since August 2017, serum samples from a further 537 external patients with confirmed or suspected inflammatory neuropathies have been received for diagnostic nodal/paranodal antibody testing by the Oxford laboratory. Clinical information was requested for all patients, including details of treatments used, and a clinician-led, subjective, overall impression of their efficacy.

### 2.2. IA Patient Cohort

The IA cohort consisted of 60 subjects (20 with CIDP, 20 with GBS, and a control group of 20 with multiple sclerosis/clinically-isolated syndrome, MS/CIS) who were selected from patients treated with IA between June 2013 and January 2018 in the University of Ulm, Department of Neurology based on the inclusion criteria outlined below. The study was reviewed by the appropriate ethics committee of the University of Ulm (approval number 20/10) and was performed in accordance with the ethical standards of the Declaration of Helsinki from 1964. Written informed consent for the sample collection was obtained from all patients participating in this study.

#### 2.2.1. CIDP

All patients with CIDP fulfilled the EFNS criteria for possible, probable, or definite CIDP, had a continuously progressive course of disease, and had previously received several cycles of steroids (*n* = 5), IVIg (*n* = 2) or both (*n* = 13), with insufficient response. Fifteen patients who had previously received IVIg showed further disease progression under IVIg therapy, therefore we opted for a new therapeutic approach with IA. In 5 patients who had never received IVIg we chose IA instead of IVIg based on our favourable clinical experience with IA in CIDP. Two patients had never been treated with prednisolone because of severe diabetes mellitus. Further treatments included azathioprine (*n* = 5), cyclophosphamide (*n* = 1), mycophenolate mofetil (*n* = 2), and methotrexate (*n* = 1). Assessment of the clinical outcome directly and 2 weeks after IA was based on the Inflammatory Neuropathy Cause and Treatment (INCAT) score [38] and the Ulmer CIDP score, which includes the INCAT, the Oxford muscle strength grading scale (Medical Research Council, MRC), and vibration sensitivity testing [33].

#### 2.2.2. GBS

All patients with GBS showed the typical clinical picture including rapidly progressive bilateral limb weakness and sensory deficits, hypo-/areflexia, electrophysiological signs of demyelination, and increased protein levels in cerebrospinal fluid. Anti-ganglioside antibodies were not tested prospectively. In contrast to CIDP and MS, IA was a first-line therapy in 4 GBS patients, and used as an escalation therapy in 9 more. In order to establish equally sized subgroups, the GBS group included 7 patients who received PLEx rather than IA. Classification of the clinical outcome (no improvement, equivocal improvement, partial improvement, large improvement) directly after the last treatment was retrospectively based on the neurological examination as documented in the medical records (discharge letter) of each patient.

#### 2.2.3. MS/CIS

All patients fulfilled the 2017 MacDonald diagnostic criteria for MS [39] or CIS. All patients treated with IA suffered from a steroid-refractory relapse, i.e., an acute relapse without complete remission after one or more cycles of high dose intravenous methylprednisolone (IVMP) therapy (at least 3 × 1000 mg). Assessment of the clinical outcome directly after the last treatment was based on the Expanded Disability Status Scale (EDSS).

### 2.3. IA Treatment

One cycle of IA consisted of five treatments on 5 consecutive days. The total plasma volume of each patient was calculated using body weight, height, and haematocrit. Two plasma volumes were processed during the first treatment, and 2.5 plasma volumes were processed during all the subsequent treatments. The Adsorber system (ADAsorb, medicap clinic GmbH, Ulrichstein, Germany) contained two regenerating protein A columns (Immunosorba, Fresenius Medical Care, Bad Homburg, Germany).

### 2.4. Sample Collection and Storage

Eluate samples were obtained during each IA treatment and buffered with bicarbonate (pH 7.0). Serum samples were obtained before and after each IA treatment. A standardized protocol for serum and eluate collection was applied as previously recommended [40]. All biosamples were stored according to the predefined standard operating procedure (SOPs) at the local biobank in Ulm at minus 80 °C within two hours. Later they were transferred for measurement on dry ice to Oxford for further analysis. 

### 2.5. Serological Analysis

Sera and eluates from the 3 patient cohorts and from control subjects were analysed using a nodal/paranodal antibody cell-based assay, paranodal, ganglioside and sulfatide ELISA, and against myelinating co-cultures. Methodological details for these experiments are given in Appendix B.

## 3. Results

### 3.1. Nodal/Paranodal Antibody (PNAb) Diagnostic Cohort

#### 3.1.1. Demographics, Clinical and Serological Characteristics

Since August 2018, serum samples from 537 different patients with confirmed or suspected inflammatory neuropathies have been received for diagnostic nodal/paranodal antibody testing by the Oxford laboratory, and we have tested a further 88 patients from our own research cohort. Overall, 42/625 patients (6.7%) were positive for nodal/paranodal antibodies (PNAb+), comprising 16 (2.6%) with NF155 specific antibodies, 1 (0.2%) with NF186 specific antibodies, 6 (1%) with pan-neurofascin antibodies, 12 (1.9%) with contactin-1 (CNTN1) antibodies and 7 (1.1%) with contactin-associated protein (Caspr1) or CNTN1/Caspr1-complex antibodies. The median age of the PNAb+ patients was 58 (range 15 to 79) and 30/42 (71.4%) were male. The initial clinical diagnosis was CIDP in 28 (66.6%), GBS in 13 (31.0%) and atypical multifocal motor neuropathy in 1 (2.4%). In one patient, the diagnosis of CIDP was subsequently revised to motor neuron disease; the diagnosis of an inflammatory neuropathy was retained at follow up in all other antibody positive cases. The remaining 583 patients were paranodal antibody negative (PNAb-negative), with clinical data available for 185 patients. The median age of the PNAb-negative patients was 62 (range 4 to 90) and 120/185 (64.9%) were male. The initial clinical diagnosis was CIDP in 100 (53.8%), combined central and peripheral demyelination in 3 (1.6%), GBS in 38 (20.4%), and multifocal motor neuropathy in 16 (8.6%). In 9/131 (6.9%) patients for whom follow up data was available, the diagnosis was subsequently revised away from that of an inflammatory neuropathy. Summary demographic and clinical details of the subgroups of apheresis treated PNAb-positive and PNAb-negative patients are given in Table 1. There was no significant difference in the median age, sex distribution, clinical diagnosis, or other serological results between the 2 subgroups. There was a non-significant trend towards more severe disease and more frequent IgG and less frequent IgM paraprotein detection in PNAb-positive patients. The frequencies of prior IVIg, steroid, PLEx and immunosuppressant use was also similar between the groups, while rituximab and IA were significantly more likely to have been used in the PNAb-positive group. PLEX aside, there was, however, no statistically significant difference in the clinician reported responses to these therapies between the 2 groups, although there was a trend to rituximab being more often judged effective in the PNAb-positive compared to PNAb-negative group.

#### 3.1.2. Physician-Reported Subjective Evaluation of Responses to Plasma Exchange or Immunoadsorption

Of the PNAb+ patients, 17 were treated with PLEx alone, 1 with IA alone, and 3 with both modalities. Protein A columns were used for three of the IA treated patients, the other (described in detail below) was treated with a GAM-peptide-ligand-based column (Globaffin, Fresenius Medical Care (UK) Ltd, Sutton-in-Ashfield, UK). Serial disability measures are available for only one other PNAb+ patient: a 68-year-old lady with a clinical diagnosis of GBS. Neither her overall neuropathy limitations score (ONLS, 12/12) nor inflammatory neuropathy Rasch-built overall disability score (iRODS, 0/48) improved following 2 cycles 5 treatments of PLEx starting on days 40 and 69 of her illness, prior to her death on day 110 from infectious complications. For all other PNab+ patients, only clinician-reported, retrospective, and subjective evaluations of response were available. None of the treating clinicians judged that either PLEx or IA had produced a subjectively “good” response in any of the PNAb+ patients. With PLEx, 5 patients (25.0%) were reported as having had a partial response, 2 (10.0%) an equivocal response, 12 (60.0%) no response, and one to have deteriorated (5.0%). With IA, 1 (25%) partial response, and 1 (25%) equivocal response were reported, with 2 patients (50%) reported as showing no response (Figure 1A,B). The proportion of PNAb+ patients subjectively judged as showing a partial or better response to PLEx (25.0%) versus IA (25.0%) was identical.

Of the PNAb-negative patients, 33 were treated with PLEx: 8 patients (24.2%) were subjectively reported as having a good response, 10 (30.3%) a partial response, 1 (3.0%) an equivocal response, 8 (24.2%) no response, and 2 (6.1%) as deteriorating. For 4 patients, the response to PLEx was not reported. Amongst the 3 ganglioside antibody positive patients, 2 were reported as having a partial response, and 1 no response, to PLEx. Apheresis, with or without IA, was significantly more likely to have been reported by treating clinicians to have produced partial or better response in the PNAb-negative patients (62.1%) compared to the PNAb+ patients (25.0%) (*p* = 0.01, Fisher’s exact test, OR 4.9 (95% CI 1.52 to 14.88) (Figure 1C,D). It should be emphasised that this is a comparison of the physicians’ subjective overall impression of response, rather than an evaluation of the true efficacy, or otherwise, of these treatments.

### 3.2. Detailed Profile of an NF155 Antibody Positive Patient Treated with Immunoadsorption

This 46-year-old male first presented to neurology in July 2019 with a 6-week history of ascending numbness and paraesthesia in his feet, then hands. He had lost the ability to run and found walking to be unsteady. On examination, power was full, but there was global areflexia with distal sensory loss to temperature, pin-prick, vibration and proprioception. His gait was broad-based and unsteady and Rhomberg’s test was positive after 20 s of eye closure. There was a postural tremor of both hands without cerebellar or extrapyramidal signs. The presentation was felt to be consistent with sensory ataxic CIDP. Neurofascin-155 antibody mediated disease was high in the differential. A positive result on the NF155 CBA and ELISA was duly returned 2 days later, at an initial titre of 1:6400. IgG4 was the dominant subclass, with IgG1 and IgG2 also represented (Figure 2A,B). CSF was acellular with an elevated protein (1.8 g/L). Nerve conduction studies showed absent median but preserved sural sensory nerve action potentials. Distal motor latencies and F-wave latencies were significantly prolonged, with slowing of intermediate motor conduction velocities. There was conduction block without temporal dispersion in the sampled peroneal nerve between the ankle and fibular head. Pulsed dexamethasone was commenced 4 days later (40mg per day for 4 days every 4 weeks for 3 cycles). There was no change in the examination findings. A progressive deterioration in symptoms and disability measures prompted a trial of IVIg (2 g/kg over 5 days) which resulted in a pompholyx-type skin rash, and no neurological benefit over the next 6 weeks. Approval was then sought for rituximab, and IA was arranged as a potential temporising measure.

Four treatment sessions of 2–2.5 plasma volumes were given on 4 consecutive days using a multiple pass, GAM-peptide-ligand-based column (Globaffin, Fresenius Medical Care Ltd, Sutton-in-Ashfield, UK). IA was effective in rapidly and substantially reducing the NF155 antibody titre (Figure 2C), but this had returned to baseline by 1 month (Figure 2D) and there was no observed clinical benefit. Rituximab was then given (1g on 2 occasions 2 weeks apart) followed by a second cycle of 5 treatments sessions of IA 1 month later. This was again associated with a rapid and substantial reduction in NF155 antibody titre, which on this occasion recovered more slowly and incompletely (Figure 2D). This more persistent suppression of antibody titres was associated with a progressive improvement in symptoms and disability, which is currently ongoing (Figure 3).

### 3.3. Demographics and Clinical Characteristics of the IA Treated Cohort

#### 3.3.1. CIDP

Details of this cohort are given in Appendix C (Table A1). Sixteen of these 20 CIDP patients have been described in a previous publication [33]. 16/20 (80%) were male. At the start of IA treatment, the cohort had a median age of 66 (range 27 to 80), and a median disease duration of 95.5 months (range 63 to 139). All had progressive disease and met the European Federation of Neurological Societies (EFNS) criteria for definite, probable, or possible CIDP [41]. 18/20 had been previously treated with corticosteroids and 14/20 with IVIg, with sub-optimal responses. Six patients were treated with at least one of azathioprine, cyclophosphamide, mycophenolate mofetil or methotrexate. Nine patients received multiple (range 2–9) cycles of IA. Five patients showed improvements in their Inflammatory Neuropathy Cause and Treatment (INCAT) disability score when assessed 2 weeks after initial IA treatment, and 8 patients showed substantial improvements (at least 10 points) in the CIDP score. 

#### 3.3.2. GBS

Details of this cohort are given in Appendix C (Table A2). 10/20 patients (50%) were male. At the start of IA or PLEx treatment, the cohort had a median age of 66 (range 31 to 89). IA was applied to 13/20 patients. IA was used as a first-line therapy in 3, as a second-line therapy (after unsuccessful treatment with IVIg) in 9, and as a third-line therapy (after both IVIg and PLEx) in 1 patient. This subgroup was supplemented with 7 patients who received PLEx, instead of IA. In these patients, PLEx was used as a first-line therapy in 6, and as a second-line therapy (after IVIg) in 1 patient. 18/20 patients received 1 cycle of IA or PLEx, and only 2 patients received 2 cycles. 4/20 (3/13 IA, 1/7 PLEx) patients showed no clinical improvement after the last treatment, 3 patients (2/13 IA, 1/7 PLEx) showed equivocal improvement, 8 patients (4/13 IA, 4/7 PLEx) showed partial improvement, and 5 patients (4/13 IA, 1/ PLEx) showed large improvement.

#### 3.3.3. MS/CIS

Details of this cohort are given in Appendix C (Table A3). 15/20 patients (75%) were female. At the start of IA treatment, the cohort had a median age of 29 (range 15 to 57). Patients were diagnosed with MS (16/20) or CIS (4/20), and had all been treated unsuccessfully with at least one cycle of high-dose intravenous methyl prednisolone (MP). 8 patients had received 2 or more cycles of high-dose IVMP. 11/20 patients showed an improvement of EDSS after the last IA treatment, while 9/20 patients did not improve.

### 3.4. Glycolipid and Nodal/Paranodal Antibodies in the IA Cohort

Pre-treatment serum samples from the IA cohort were tested for sulfatide and GM1- and GQ1b-ganglioside IgG antibodies by ELISA. None of these sera were positive on these assays. Serum samples taken pre and post-treatment, as well as first treatment session eluates from the IA cohort (20 CIDP, 20 GBS and 20 MS/CIS patients), were tested by both cell-based assay (CBA) and ELISA for antibodies to nodal (neurofascin-186) and paranodal (neurofascin-155, contactin-1 and Caspr) cell adhesion molecules. None of the sera were positive on either assay. One eluate from the MS/CIS cohort (patient 09) was positive on the neurofascin-155 CBA (blind scored as ‘2+’ at 1:100, end-point titre 1:200, Figure 4A) (For scoring method see Section A.1). The sole detected subclass was IgG1. The corresponding pre-treatment serum was negative for NF155 antibodies at 1:100, the standard screening titre for this assay, but scored 3+ when repeated at 1:20. Two further eluates, one from the MS/CIS cohort and one from the CIP cohort, also produced faint membrane binding (1+) on the neurofascin-155 CBA that was not sufficient to be called positive at 1:100. Repeat testing of these eluates at 1:20 increased the signal to 2+ and 3+ respectively. However, this titre is below the usual positivity cut-off for this assay, and no signal was produced with any of the IgG subclass-specific secondary antibodies. All of these eluates were negative on the neurofascin-155 ELISA and negative for all other antigens by both CBA (including neurofascin-186, Figure 4B) and ELISA (results not shown). 

### 3.5. Screening the IA Eluates for Novel Antibodies Using Myelinating Co-Cultures

In this experiment, eluates from the first treatment session of each IA cohort were compared with purified IgG from the serum of 22 healthy control volunteers (gratefully received from A/Prof Sarosh Irani, University of Oxford) isolated by Protein G purification. Serum was not available in sufficient quantities from PNAb cohort to purify IgG and these samples were therefore not tested in this experiment. IgG from IA eluates (1:50 dilution) and protein G purification (1:12.5 dilution) were applied to myelinated human sensory neuron cultures in a 96 well, flat-bottom imaging plate format enabling high-throughput staining and imaging. The mean IgG concentration after dilution was not significantly different between the groups (One-Way ANOVA: F(3,78) = 1.500, *p* = 0.2211) (Figure 5A). Out of 82 samples tested, 1 CIDP (patient 11), 1 GBS (patient 07, who was also concurrently identified as HIV positive, see Section B.2 for further detail) and 1 MS/CIS (patient 13) sample were scored as ‘positive’ for either axonal, glial or nodal IgG deposition by an observer blinded to the patient group; a further 1 MS/CIS patient sample with weak IgG labelling was marked ‘equivocal’. All 4 of these sera and IA eluates were negative on the glycolipid and paranodal antibody assays, as above. Pre-treatment serum from MS patient 13 was also negative on our in-house live CBAs for aquapaorin-4 and MOG antibodies. Neither of the MS/CIS eluates which produced a weak signal on the neurofascin-155 CBA were positive on the co-culture assay.

Serum samples taken pre- and post-IA from the four candidate patients (1:50 dilution) were further validated on myelinated cultures plated on 13 mm coverslips with careful attention paid to media changes and washing steps. Strong IgG deposition aligned with neurofilament positive axons was observed in the serum and IA eluate of the GBS (patient 07) (Figure 5B) and CIDP (patient 11) (Figure 5C) patients. We confirmed nodal reactive IgG in the serum and IA eluate of one MS patient (Figure 5D and Appendix A), which was absent from post-IA serum. The post-treatment follow-up serum from the CIDP (patient 11) patient was negative for any IgG reactivity (Figure 5E). No IgG reactivity was observed in the serum or eluate of the MS/CIS patient 13 previously marked as equivocal, confirming this as a false positive. Clinical vignettes describing the patients with IgG deposition on co-cultures are given in Appendix C.

## 4. Discussion

In the PNAb cohort, we found that PLEx or IA were more often subjectively judged to have been effective in seronegative cases, and that in contrast, detection of at least one of the known nodal/paranodal antibodies in patients with inflammatory neuropathies was not associated with clinicians perceiving a positive response to either treatment. The proportion of PNAb-negative patients judged to have had a partial or better response (62.1%) was similar to the proportion of patients judged to have had a partial or better response in the IA cohort (52.5% overall), all of whom were also negative for known nodal/paranodal antibodies. We emphasise that the evaluation of the PNAb cohort is limited by the retrospective and subjective nature of the patient assessment. In addition, the small number of cases precludes us from reaching any conclusions regarding the objective benefits of one treatment modality compared to the other in this setting. In addition, improvement in neurological symptoms following IA/PLEX may occur after a delay, which may not be reflected in the immediate judgement of the treating physician. Blinding, randomisation, standardised follow up, as well as a control group to judge the natural history of these heterogeneous diseases, are required for a definitive evaluation of apheresis treatment efficacy in these patient groups. However, it is notable that treating physicians were less likely to think that apheresis had been effective in PNAb-positive patients.

Why seropositive patients were rarely assessed to have responded positively to either IA or PLEx is unclear. Our close monitoring of a prospectively-identified neurofascin-155 positive individual showed that while IA given as a mono-therapy was able to effectively reduce antibody titres, levels quickly rebounded and reached pre-treatment levels inside 4 weeks. This transient serological effect was not sufficient to reduce disability. More prolonged suppression of antibody titres, with frequent apheresis cycles or adjuvant therapies, may therefore be required for effective treatment in such cases.

Rituximab has previously been suggested as an effective treatment for paranodal antibody positive patients [42,43], but may take several weeks (or even months) to produce benefit. In this case, a second cycle of IA, 4 weeks after a course of rituximab, produced a more persistent suppression of antibody titres, which was associated with clinical improvement. The extent to which IA contributed to this effect is unclear. Theoretically, the more rapid action of IA might be complementary to the delayed but more sustained effects of rituximab. Whether this combination of treatment offers significant benefit over rituximab alone requires further investigation.

Retrospective analysis of serum samples from 60 IA-treated patients failed to identify any individuals who would have been classified as positive on routine diagnostic testing for previously described nodal/paranodal and glycolipid antibodies. A small number of first-treatment IA eluates did produce a low-level signal on the neurofascin-155 CBA. Whilst the diagnostic importance of low-titre, non-IgG4 results has been doubted [44], a pathogenic role for these antibodies cannot be ruled out.

The apparently better response of seronegative patients to apheresis, particularly IA, has several possible explanations. One is that these differences simply reflect variation in the disease characteristics and natural progression of seropositive versus seronegative inflammatory neuropathies: Overall, seropositive patients tend to have more severe, aggressive disease that is refractory to treatment [30,31,32]. Conversely, less severely affected, seronegative, patients may be more likely to have a monophasic disease course and stabilise or improve, independent of any particular therapy. Indeed, the median peak disability, measured by nadir modified Rankin score (mRs), of apheresis-treated PNAb+ patients in our series was higher, albeit non-significantly, than that of the apheresis-treated seronegative group (median nadir mRs 5 v 4, *p* = 0.1, Mann-Witney test, Table 1), although there was no significant difference in the use of, or clinician evaluated response to, other treatment modalities. There was also no significant difference in the proportion of patients initially diagnosed as GBS (28.6% and 30.3%, *p* > 0.99) compared to CIDP (66.7% and 54.5%, *p* = 0.41) in the PNAb+ and PNAb-negative groups, respectively (Fisher’s exact test, Table 1). However, this does not exclude the possibility that patients in the seronegative group may often have a shorter disease course, with less irreversible axonal degeneration.

Another explanation for perceived apheresis efficacy in seronegative patients is the presence of antibodies below the threshold for positive detection on diagnostic testing, leading to a correspondingly slower rebound in titres following PLEx/IA and a more sustained suppression of antibody levels. A further possibility is that the response to IA in diagnostically seronegative patients is due to the therapeutic removal of as-yet uncharacterised, pathologically relevant antibodies in these patient groups. We therefore tested for further nerve-related antigens by screening eluates from the IA cohort against myelinating co-cultures. Three positive IA eluate samples were identified in the original 96-well co-culture screen and were further validated in a larger 24-well format, confirming similar binding patterns. IgG from one GBS patient co-localised with NF200 suggesting an axonal antigen. One CIDP patient serum and IA eluate showed IgG binding that aligned with NF200-positive axons but may also reflect deposition on non-myelinating Schwann cells.

One patient’s serum and IA eluate from the MS/CIS group revealed nodal specific IgG binding. The presence of antibodies against nodal antigens such as neurofascin, has precedence in MS, and although uncommon, is more predominant in chronic progressive forms of the disease [45]. However, this sample was negative for antibodies against both the glial/paranodal and nodal/axonal isoforms of neurofascin (NF155 and NF186, respectively). The original focus on peripheral neuropathies led us to use a sensory neuron system for the myelinating cultures. Nevertheless, multiple peripheral nerve antigens are also found in the CNS (and vice versa), including NF155, CNTN1 and the ganglioside GM1. Therefore, it is quite feasible for the unknown antigen targeted by IgG in this CIS/MS patient to be mutually expressed in the peripheral and central nervous systems (CNS). Other autoantibodies against nerve and glial structures in the CNS including myelin basic protein, myelin-associated lipids, contactin-2, and KIR4.1 are among those proposed in MS patients [46]; however, their presence may not be specific to the disease [47]. For this reason, the inclusion of MS/CIS patients as a control group is potentially problematic. However, as patients with non-autoimmune neurological disease essentially never receive apheresis treatment, the inclusion of this group was a pragmatic way to obtain non-neuropathy IA eluates for use in our unbiased screening assays. With some similarity to the discovery of nodal/paranodal antibodies in chronic neuropathies, MS has recently been separated from other distinct, serologically-defined disorders, characterised by the presence of aquaporin-4 or myelin oligodendrocyte glycoprotein (MOG) directed autoantibodies. Whether the nodal antigen targeted by antibodies in this MS patient has a pathogenic role and might similarly define a non-MS disease entity is currently unknown. Further investigation using brain tissue may help elucidate the antigen target, pathological potential, and clinical relevance. Unfortunately, purified Ig/eluate was not available from the PNAb-negative apheresis cohort, and it is possible that novel autoantibodies are also present in some of these patients.

The two patients for whom follow-up samples were available (CIDP and MS/CIS) had no detectable IgG labelling in their serum after IA compared to pre-treatment. Thus, IA is effective at removing both established and potentially novel pathogenic autoreactive IgG from the circulation. Follow-up serum samples at later time points will help correlate any changes in disease progress with antibody titres.

Development of myelinated hiPSC-derived neuronal cultures in a 96-well format allowed for efficient simultaneous screening of >80 IgG eluates from patients and controls. The benefits of using live cultures for screening are the presence of complex structures including nodes of Ranvier, paranodal and juxtaparanodal regions, and compact myelin internodes, that provide an unbiased substrate for antibody screening against nerve-related antigens in their native conformation. IgG binding patterns ranged from broad axonal coverage to focal nodal localisation, reflecting morphologically distinct antigens. Images were acquired by an experienced observer who was blind to the sample identity. Although time-consuming, acquisition in such a supervised manner aids the detection of localised signals, such as the node-specific labelling identified in one MS/CIS patient.

A single sample that was marked as ‘equivocal’ on the 96-well assay was subsequently confirmed as negative. The minimal occurrence of non-specific IgG labelling in the 96 well format may reflect a lower washing efficiency in the smaller volume of the 96-well plate. Nevertheless, no healthy control samples were identified as positive in the screen, suggesting the cultures are useful as a selective substrate for nerve-targeted autoantibodies.

IA is rarely performed on healthy subjects; therefore control IgG were prepared from the sera of healthy volunteers by protein G purification. IgG concentrations in the healthy samples were normalised to the patient IA eluates such that the mean IgG concentrations were not significantly different, however the variation within each group was maintained in order to reflect the original sample. The detection of specific signals in both the serum and IA eluate of each of the three positive patients suggests that a uniform dilution of 1:50 is sufficient for antibody screening within IA eluates. We cannot, however, exclude the possibility of further antibodies below the level of detection. In summary, our findings of nerve antigen reactive antibodies in three ‘seronegative’ neurological patients suggest the utility of an unbiased screening system such as we have described here for the myelinating co-cultures. The development of equivalent cultures containing CNS antigens and cell-types may be of further benefit to relevant MS cases.

## 5. Conclusions

Currently available serological tests do not unambiguously identify patients who are likely to respond to IA or PLEx. In patients with nodal/paranodal antibody associated neuropathies, frequent plasmapheresis and/or additional therapies may be required to produce an acceptable level and duration of clinical improvement. Prospective longitudinal studies involving standardized and validated outcome measures, with serial monitoring of auto-antibodies, are needed to optimise apheresis treatment regimens and accurately assess efficacy.

## Figures and Tables

**Figure 1 jcm-09-02025-f001:**
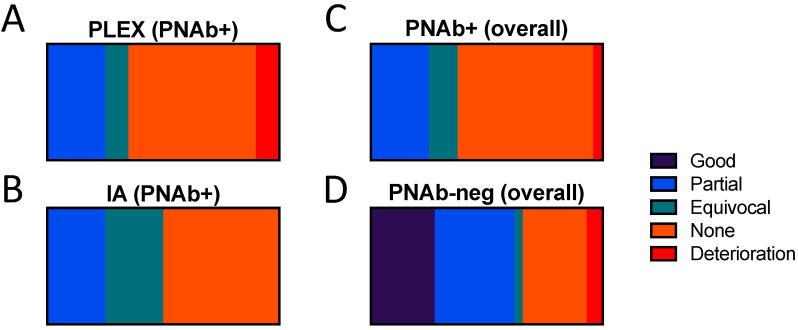
Physician-reported subjective evaluation of response to plasma exchange or immuno-adsorption in paranodal antibody positive and negative patients. Paranodal antibody positive patients treated with (**A**) plasma exchange (*n* = 17), (**B**) immunoadsorption (*n* = 4), or (**C**) either modality (*n* = 21), compared to (**D**) paranodal antibody negative patients (*n* = 33) (all treated with plasma exchange).

**Figure 2 jcm-09-02025-f002:**
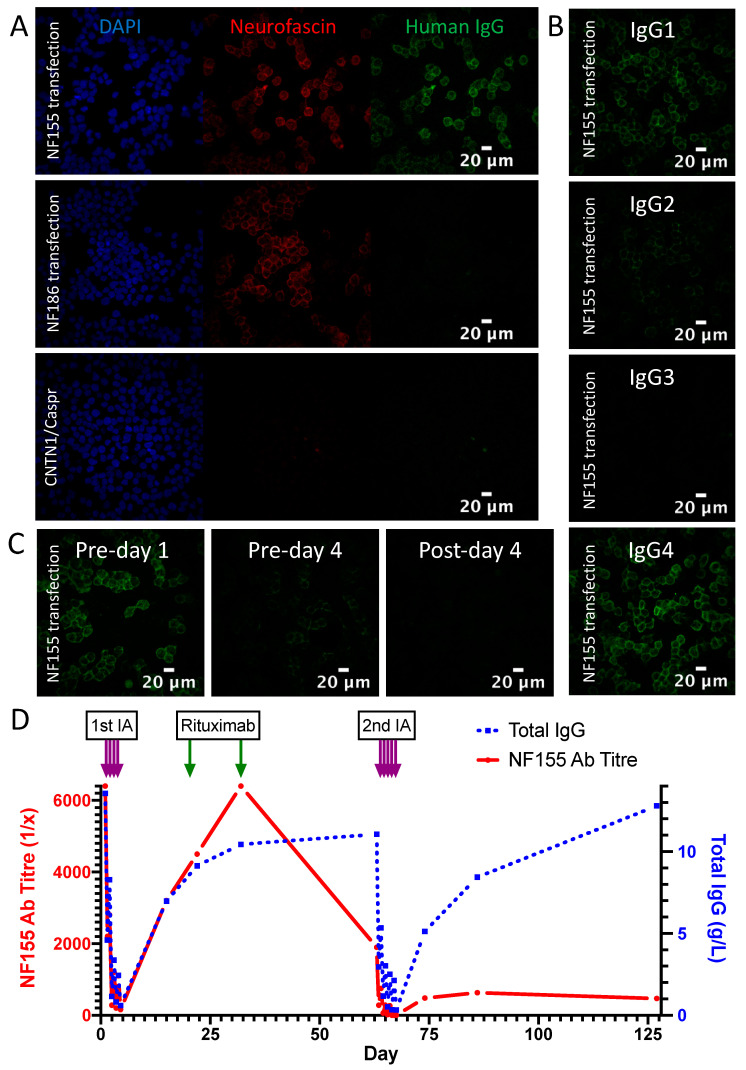
Serological results of NF155 antibody positive patient at baseline and during IA treatment. (**A**) Serum contains IgG (green) which binds to the cell membrane of NF155-transfected HEK293T cells, and co-localises with a commercial pan-neurofascin antibody (red). No signal is seen with NF186 or CNTN1/Caspr1-transfected cells. (**B**) The predominant IgG subclass of the NF155 antibodies is IgG4, with IgG1>IgG2 also represented. (**C**) The antibody signal intensity at 1:100 before, during and immediately after the first cycle of IA shows a progressive decline. (**D**) NF155 antibody titre (red) and total IgG levels (blue) over 2 cycles of IA, before and after rituximab.

**Figure 3 jcm-09-02025-f003:**
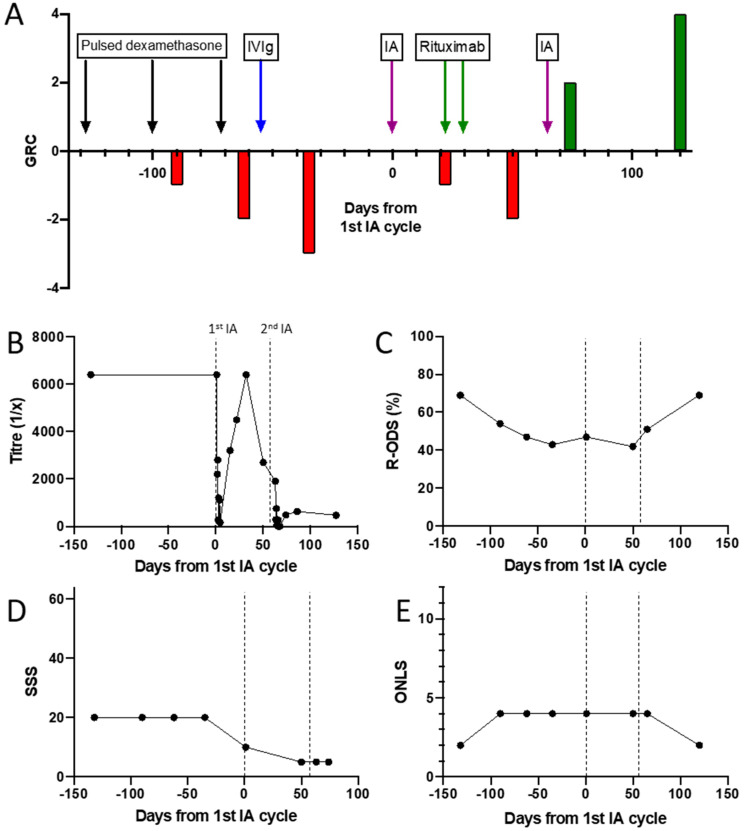
Antibody titres and outcome/disability measures during treatment of a patient with an NF155-antibody-mediated neuropathy. (**A**) Patient global rating of change after treatment with dexamethasone, IVIg, IA and rituximab. (**B**) NF155 antibody titre. (**C**) Inflammatory neuropathy Rasch-built Overall Disability Score. (**D**) Sensory sum score. (**E**) Overall neuropathy limitations score.

**Figure 4 jcm-09-02025-f004:**
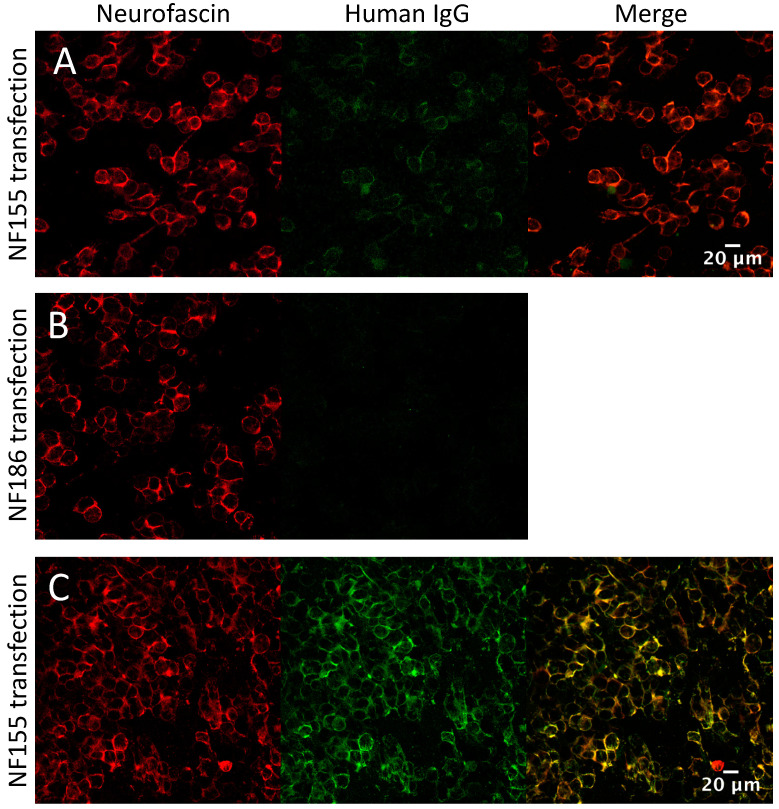
Nodal/paranodal cell-based assays. (**A**) MS/CIS eluate weakly positive on the neurofascin-155 CBA at 1:100 (Score 2+, end-point titre 1:200) and (**B**) negative on the neurofascin-186 CBA. (**C**) Strong positive at 1:100 (Score 4+, end-point titre 1:3200) from the antibody positive CIDP cohort shown for comparison.

**Figure 5 jcm-09-02025-f005:**
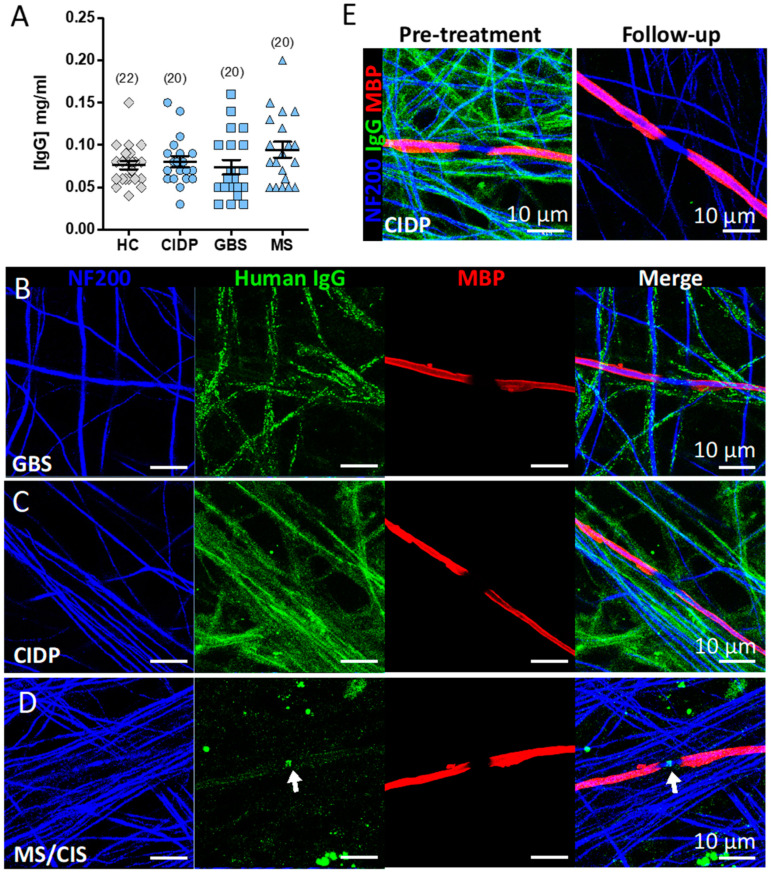
IgG deposition in myelinated co-cultures. (**A**) IgG concentration of dilution-adjusted eluates used for screening on myelinated cultures. (**B**–**D**) Immunofluorescence images of IgG binding patterns in myelinating co-cultures of IA eluates (1:50) from three patients with neurological disease identified in the screening assay: B) GBS (patient 07), C) CIDP (patient 11), and D) MS/CIS (patient 13) (arrow indicates IgG deposition at the node of Ranvier). (**E**) IgG labelling in myelinated co-cultures of serum (1:50) sampled from the CIDP (patient 11) before (Pre-treatment) and after IA (Follow-up). Note all IgG immunoreactivity is lost at follow-up. NF200, neurofilament 200; MBP, myelin basic protein.

**Table 1 jcm-09-02025-t001:** Summary characteristics of apheresis treated patients from the PNAb cohort.

	PNAb Positive (*n* = 21)	PNAb Negative (*n* = 33)	Significance (PNAb+ v PNAb-neg)		
Age: median, (range)	58 (35–79)	62 (5–90)	ns	*p* = 0.94	Mann-Whitney
Male sex: *n*, (%)	16 (76.2%)	23 (69.7%)	ns	*p* = 0.76	Fisher’s exact
Initial clinical diagnosis:					
● GBS: *n* (%)	6 (28.6%)	10 (30.3%)	ns	*p* > 0.99	Fisher’s exact (GBS or not)
● CIDP: *n* (%)	14 (66.7%)	18 (54.5%)	ns	*p* = 0.41	Fisher’s exact (CIDP or not)
● Other: *n* (%)	1 (4.7%)	5 (15.1%)	ns	*p* = 0.39	Fisher’s exact (Other or not)
Peak severity/nadir mRs (median, range)	5 (2–6)	4 (2–5)	ns	*p* = 0.10	Mann-Whitney
Other serology: *n*/*n* (%)					
Any ganglioside Ab	1/16 (6.3%)	3/18 (16.7%)	ns	*p* = 0.60	Fisher’s exact
● GM1	1/16 (6.3%)	2/18 (11.1%)	ns	*p* > 0.99	Fisher’s exact
● GQ1b	0/16	1/18 (5.6%)	ns	*p* > 0.99	Fisher’s exact
MAG	0/4	1/8 (12.5%)	ns	*p* > 0.99	Fisher’s exact
Paraprotein	2/17 (11.8%)	6/26 (23.1%)	ns	*p* = 0.45	Fisher’s exact
● IgM	0/17	5/26 (19.2%)	ns	*p* = 0.14	Fisher’s exact
● IgG	2/17 (11.8%)	1/26 (3.8%)	ns	*p* = 0.55	Fisher’s exact
**Treatment** **% treated (% of those judged to have good response)**			**Difference in** **proportion treated/proportion with good response**		
IVIg	90.5 (5.3%)	87.9% (3.4%)	ns/ns	*p* > 0.99/*p* > 0.99	Fisher’s exact
Steroids	85.7% (0)	75.8% (8%)	ns/ns	*p* = 0.50/*p* = 0.50	Fisher’s exact
PLEx	95.2% (0)	100% (24.2%)	ns/*	*p* = 0.39/**p* = 0.01	Fisher’s exact
IA	19% (0)	0 (0)	***/ns	****p* < 0.001/ *p* > 0.99	Fisher’s exact
Rituximab	66.7% (64.3%)	18.2% (16.7%)	***/ns	****p* < 0.002/*p* = 0.14	Fisher’s exact
Other immuno-suppression	33.3% (28.6%)	24.2% (12.5%)	ns/ns	*p* = 0.54/*p* = 0.47	Fisher’s exact

GBS, Guillain-Barré syndrome; CIDP, chronic inflammatory demyelinating polyneuropathy; GM1, monosialoganglioside GM1; GQ1b, tetrasialoganglioside GQ1b; MAG, myelin associated glycoprotein; IVIg, intravenous immunoglobulin; PLEx, plasma exchange; IA, immunoadsorption, * and ***, indicate statistical significance.

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
