# Peer review of "Immunoadsorption and Plasma Exchange in Seropositive and Seronegative Immune-Mediated Neuropathies"

_jcm, 2020, doi:10.3390/jcm9072025_

Round 1

Reviewer 1 Report

See attached peer review summary

Author Response

Manuscript ID: jcm-823524

Response to reviewer 1

The authors have performed a retrospective analysis of cohorts of patients from Ulm, Germany and Oxford to identify serological markers, such as antibodies to peripheral nerve antigens, that correlate with outcome following treatment with either plasma exchange (PLEX) or immunoadsorption (IA).

The major limitations from this study are due to the retrospective data collection in the Oxford PNAb cohort. No results are presented from myelinating co-cultures from the Oxford PNAb cohort, presumably because this was a retrospective study, making it difficult to compare the results with the Ulm cohort of inflammatory neuropathies.

In addition, the response to PLEX/IA in the PNAb Oxford cohort was clinician-reported rather than from well-validated clinical scores, which significantly limits the ability to draw significance from the difference in outcomes between PNAb positive and negative patients.

Thank you for your comments. We have made a point by point response to these in blue text, as below. Please note that the line numbers refer to the revised manuscript when viewed with tracked changes visible and showing “All markup”. The line numbers will change if the document is viewed without this markup visible.

We acknowledge the limitations of the clinician-reported assessments of response to PLEX/IA in the PNAb Oxford cohort. Throughout the manuscript we state that we are evaluating the “judged” or “[clinician] reported” or “physician perceived” response, but have now further emphasised the subjective nature of this assessment (see L20, L109, L126, L220, L230/5) and further clarified we are comparing the physicians’ impression of response rather than making a genuine assessment of treatment efficacy (L254-256)

 “It should also be emphasised this is a comparison of the physicians’ overall impression of response, rather than an evaluation of the true efficacy or otherwise of these treatments.“

We agree that we are limited in our ability to assess outcomes in this cohort, or assess whether these are significantly different.  This is not the message we intend to communicate from these results, and we have further emphasised this. L383 to 387 of the original discussion stated:

“The nature of this study prevents a genuine evaluation of whether these treatments are truly effective (or not) in these patient groups. Major limitations include the retrospective and, in many cases, subjective nature of the assessment of benefit, the lack of blinding, randomisation, and standardised follow up, and the absence of a control group to judge the natural history of these heterogenous diseases.”

We have modified this statement for clarity in the revision (L409-416):

“We emphasise that the evaluation of the PNAb cohort is limited by the retrospective and subjective nature of the patient assessment. In addition, the small number of cases precludes us from reaching any conclusions regarding the objective benefits of one treatment modality compared to the other in this setting. In addition, improvement in neurological symptoms following IA/PLEX may occur after a delay, which may not be reflected in the immediate judgement of the treating physician. Blinding, randomisation, standardised follow up, as well as a control group to judge the natural history of these heterogeneous diseases, are required for a definitive evaluation of apheresis treatment efficacy in these patient groups.”

However, we maintain that treating physicians are less likely to rate PLEx or IA as effective (when assessed in a global and subjective way) in PNAb positive compared to PNAb negative subjects and that this is important for clinical decision making (a response to PLEx or IA doesn’t necessarily mean the patient is paranodal antibody positive, and vice versa) and we have stated this directly (L416-418):

“However, it is notable that treating physicians were less likely to think that apheresis had been effective in PNAb-positive patients.”

Overall, the clinical data is well presented and the conclusions reached are reasonable. However, major revision is needed that before this manuscript can be accepted for publication.

I have the following recommendations to revise the manuscript:

  1. Section 2.1.2 - clinical summary characteristics of patients with GBS - it would be helpful to include results of antibodies to ganglioside if available. This could be included in Table 2 in the Appendix.

Ganglioside antibodies are not routinely done in the Ulm clinic, since they imply no therapeutic consequence for patients with GBS. We added this information to the text (L157):

“Anti-ganglioside antibodies were not tested prospectively.”

We have however now included the results of testing for IgG GM1, GQ1b and sulfatide antibodies in the IA cohort, that was performed specifically for this study, in section 3.4 (L336-339):

“3.4 Glycolipid and nodal/paranodal antibodies in the IA cohort

Pre-treatment serum samples from the IA cohort were tested for sulfatide and GM1- and GQ1b-ganglioside IgG antibodies by ELISA. None of these sera were positive on these assays (results not shown).”

  1. Section 2.1.3 - I am not clear why patients with MS/CIS were included in this study on inflammatory peripheral neuropathies except to provide a CNS immunological disease control. The rationale for why these patients were included in the study should be addressed.

We have now stated that these patients were a control group (L26, L128).

The myelinating co-cultures were sensory neurones from human induced pluripotent stem cells rat Schwann cells - ie not CNS antigens. In addition, the significance of the single MS patient with IgG deposition at node of Ranvier is not addressed.

Shared antigenicity between the peripheral nerve and brain is common, for example NF155 reactivity is found in the cerebellum where it is thought to underlie the ataxia in NF155+ve CIDP (PMID: 24523485). Therefore cross-reactivity of certain CNS autoantibodies with our nerve cultures is not wholly unexpected. However, without a positive identification of this particular nodal antigen (this sample was negative on nodal/paranodal antibody diagnostic testing) it is premature to make any conclusions about potential pathogenicity. To reflect these factors, we have added the following to the relevant sections of the discussion (L492-496):

“The original focus on peripheral neuropathies led us to use a sensory neuron system for the myelinating cultures. Nevertheless, multiple nerve antigens are also found in the CNS (and vice versa), including NF155, CNTN1 and the ganglioside GM1, and therefore it is quite feasible for the unknown antigen targeted by IgG in this CIS/MS patient to be mutually expressed in the peripheral and central nervous systems (CNS).”

L506-8: “Further investigation using brain tissue may help elucidate the antigen target and potential pathology.”

  1. Section 3.3 - were other neurological disease controls (eg ALS, non-inflammatory neuropathies etc) studied in the co-cultures?

No – the only control groups in this study were healthy volunteers and MS/CIS patients. The MS/CIS group was a pragmatic choice for obtaining non-neuropathy IA eluates for testing, which would not be available from non-inflammatory neurological patients. There are of course certain limitations and potential confounds which we have now expanded upon.

L493: “multiple peripheral nerve antigens are also found in the CNS (and vice versa)

L498-502: “For this reason, the inclusion of MS/CIS patients as a control group is potentially problematic. However, as patients with non-autoimmune neurological disease essentially never receive apheresis treatment, the inclusion of this group was a pragmatic way to obtain non-neuropathy IA eluates for use in our unbiased screening assays.”

  1. Section 3.3 - the HIV status of GBS-07 should be highlighted as well as referencing the clinical vignette in the Appendix.

We have now indicated this as requested (L370, L785-788) (see below).

  1. Section 3.5.1 - it would be useful to summarise the demographics, clinical and serological characteristics of the PNAb+ and PNAb-ve patients in a table to be added to the Appendix.

We have now added a table detailing the basic demographic features, diagnoses, other treatments used (and physician assessed responses to these) to Table 1 in the main manuscript (L259).  

  1. Section 3.5.2 - Did any of the PNAb patients have RODS or ONLS scores before and after PLEX/IA treatment? This would be useful to include as otherwise the data is highly subjective and retrospective.

RODS / ONLS scores are available for one other PNAb+ patient before and after PLEX treatment, and this information has been added (now re-ordered as section 3.1.2, L225-229). As above, we again acknowledge the subjective and retrospective nature of the other assessments (L230).

I do not think it is sensible to report a clinically significant statistical difference in outcomes following PLEX/IA between PNAb+ and PNAB-ve patients given the limitations of the retrospective data.

As above, we agree that we cannot report a difference in clinical outcome, or true efficacy. Instead we provide the statistic as a measure of the difference in physicians’ impression of response, which is nevertheless important when deciding on further treatment options and diagnostic antibody testing. This has been clarified in the following amended sentence (L254-56):

It should also be emphasised this is a comparison of the physicians overall impression of response, rather than an evaluation of the true efficacy or otherwise of these treatments. “

We also re-state in the conclusion (L578) that:

“Currently available serological tests do not unambiguously identify patients who are likely to respond to IA or PLEx.”

  1. Section 3.5.2. - I presume that IgG was not available to study in myelinating co-cultures from PNAb-ve patients as this was a retrospective study. In contrast IgG samples were available from the CIDP and GBS cohorts from Ulm, Germany. This should be explicitly stated in the manuscript as the myelinating co-culture identified IgG deposition in a GBS and a CIDP patient from Ulm and it is possible that PNAb-ve patients in the Oxford cohort may also have IgG deposition on this test.

Correct. We have added a statement to section 3.5 (L364-66) to confirm that purified IgG / IA eluate was not available for the PNAb cohort:

“Serum was not available in sufficient quantities from PNAb cohort to purify IgG and these samples were therefore not tested in this experiment.”

We have also added a comment in the discussion about the possibility of other antibodies in the PNAb-negative group (L508):

“Unfortunately, purified Ig/eluate was not available from the PNAb-negative apheresis cohort, and it is possible that novel autoantibodies are also present in some of these patients.”

  1. Section 4. Discussion - I found the discussion hard to follow. For example, the opening statement is ambiguous and needs to be revised as there were two separate patient cohorts in the study. Does this statement applies to the PNAb diagnostic Oxford cohort study or does it also apply to the cohort from Ulm? I would recommend that the Discussion is re-written starting with review of the results from the sixty patients from Ulm before addressing the results from the Oxford PNAb cohort and the methodological limitations of the study, specifically the lack of myelinating co-culture data from the Oxford PNAb cohort.

We agree the discussion could be clearer and have re-written accordingly. We have removed the ambiguous statement and now start with a discussion of the retrospective analysis of the Oxford PNAb cohort including the limitations of this approach. This then leads onto the investigation of the serology of a defined IA cohort in which we could assess removal of any reactive IgG in eluates, as well as pre- and post-IA serum. We end with a discussion of the myelinated co-cultures and the potential benefits of extending such an unbiased assay to screen further cohorts of PNAb-negative patients for novel autoantibodies.

  1. B2 GBS-07 - was this GBS in the context of HIV seroconversion of GBS in patent with established HIV infection?

GBS was diagnosed in the context of the first diagnosis of HIV. This may well have been in the context of seroconversion, but we cannot definitely prove this either way We have clarified this in the text as follows (L370):

“[…]1 GBS (patient 07, who was also concurrently identified as HIV positive, see Appendix B2 for further detail).”

We also added this information to Appendix 3.2 (L785-788):

 “A subsequent serological HIV test was positive, initially showing 376000 HIV RNA copies per ml. This confirmed a new diagnosis of HIV infection, and raises the possibility that this gentleman’s GBS was associated with HIV seroconversion. However, in the absence of serial serological testing, we cannot confirm this unequivocally.

Reviewer 2 Report

In this study the authors retrospectively evaluated the serological status of patients with different immunomediated diseases (GBS, CIDP, CIS/Multiple sclerosis) and reviewed the response to IA or PLEx in a separate cohort of neuropathies, comparing the outcome between seropositive and seronegative patients.  

Although interesting, the study shows several limitations and biases with regard to methods, and statistics is really simple. 

The following issues require revisions.

  1. Introduction: the authors should add a sentence on the use of IA or PLEx in MS/CIS, due to the inclusion of patients affected from this condition in the study.
  2. Esperimental section: the authors “randomly” selected 60 patients from patients treated with IA between 2013 and 2018. In my opinion, the authors should have selected patients based on defined clinical inclusion criteria (for example, only patients failing a first-line therapy), making the groups more homogeneous.
  3. As regard to the clinical outcome in GBS group, the authors should better explain the “time window” established to verify the treatment response, and specify if the outcome is only anamnestic (referred from the patient) or based on clinical examination.
  4. For the sake of completeness, it would be better adding a column with DMT (disease modyfing therapy) in the table of MS patients.
  5. In my opinion, the direct comparison between PNAB+ and PNAB- patients has several important limitations. Patients were not stratified for disease severity or for previous therapies (for example immunosoppressive treatments, rituximab?). It is not clear if the outcome is defined on the basis of a neurological examination or if it is only subjective (referred from the patient). Due to the retrospective nature of the study, the small sample and the above mentioned limitations, I think that is really difficult to draw conclusions from these data. 

Author Response

Manuscript ID: jcm-823524

Response to reviewer 2

In this study the authors retrospectively evaluated the serological status of patients with different immunomediated diseases (GBS, CIDP, CIS/Multiple sclerosis) and reviewed the response to IA or PLEx in a separate cohort of neuropathies, comparing the outcome between seropositive and seronegative patients.  

Although interesting, the study shows several limitations and biases with regard to methods, and statistics is really simple. 

Thank you for your comments. We have made a point by point response to these in blue text, as below. Please note that the line numbers refer to the revised manuscript when viewed with tracked changes visible and showing “All markup”. The line numbers will change if the document is viewed without this markup visible.

The following issues require revisions.

  1. Introduction: the authors should add a sentence on the use of IA or PLEx in MS/CIS, due to the inclusion of patients affected from this condition in the study.

We have now done this (L74/76):

“There is also some evidence that apheresis can improve recovery from multiple sclerosis relapses, and these approaches are often used after inadequate responses to corticosteroids [24,25].”

  1. Experimental section: the authors “randomly” selected 60 patients from patients treated with IA between 2013 and 2018. In my opinion, the authors should have selected patients based on defined clinical inclusion criteria (for example, only patients failing a first-line therapy), making the groups more homogeneous.

We agree that the term “randomly” is misleading, since in fact we applied such inclusion criteria as outlined under 2.2 for each cohort. We therefore changed the wording as follows (L128):

“The IA cohort consisted of 60 subjects (20 with CIDP, 20 with GBS, and a control group of 20 with MS/CIS) who were selected from patients treated with IA between June 2013 and January 2018 in the University of Ulm, Department of Neurology based on the inclusion criteria outlined below.”

The inclusion criteria are described in the respective sections:

- CIDP (L136): “All patients with CIDP fulfilled the EFNS criteria for possible, probable, or definite CIDP, had a continuously progressive course of disease, and had previously received several cycles of steroids (n=5), IVIg (n=2) or both (n=13), with insufficient response.”

- GBS (L155): All patients with GBS showed the typical clinical picture including rapidly progressive bilateral limb weakness and sensory deficits, hypo-/areflexia, electrophysiological signs of demyelination, and increased protein levels in cerebrospinal fluid. In contrast to CIDP and MS, IA was a first-line therapy in 4 GBS patients, and used as an escalation therapy in 9 more.”

- MS (L165): “All patients fulfilled the 2017 MacDonald diagnostic criteria for MS [39] or CIS. All patients treated with IA suffered from a steroid-refractory relapse, i.e., an acute relapse without complete remission after one or more cycles of high dose intravenous methylprednisolone (IVMP) therapy (at least 3 x 1000 mg).”

  1. As regard to the clinical outcome in GBS group, the authors should better explain the “time window” established to verify the treatment response, and specify if the outcome is only anamnestic (referred from the patient) or based on clinical examination.

The categories used for the outcome for GBS are based on the neurological examination directly after the last treatment as documented in the discharge letters compared to the status pre-treatment. We have added the respective information to the manuscript (L160):

 “Classification of the clinical outcome (no improvement, equivocal improvement, partial improvement, large improvement) directly after the last treatment was retrospectively based on the neurological examination as documented in the medical records (discharge letter) of each patient.”

We acknowledge that improvement in neurological symptoms following IA/PLEX may occur after a delay, and that this constitutes a limitation of the retrospective nature of this study. Therefore, we highlighted this aspect in the discussion section (L412):

“In addition, improvement in neurological symptoms following IA/PLEX may occur after a delay, which may not be reflected in the immediate judgement of the treating physician.”

  1. For the sake of completeness, it would be better adding a column with DMT (disease modyfing therapy) in the table of MS patients.

We agree that this information is important and added DMT to Appendix B in Table B3 (l758).

  1. In my opinion, the direct comparison between PNAB+ and PNAB- patients has several important limitations.
  2. Patients were not stratified for disease severity or for previous therapies (for example immunosoppressive treatments, rituximab?).

Correct. These patients were an unselected cohort who were referred for diagnostic paranodal antibody testing; they were divided purely based on antibody positivity (PNAb+) or negativity (PNAb-) in diagnostic assays. We have now added an extra table to compare the basic clinical features and previous therapies of these serologically defined groups (Table 1, L259), and have added (L211-219):

“There was no significant difference in the median age, sex distribution, clinical diagnosis, or other serological results between the 2 groups. There was a non-significant trend towards more severe disease and more frequent IgG and less frequent IgM paraprotein detection in PNAb-positive patients. The frequencies of prior IVIg, steroid, PLEx and immunosuppresant use was also similar between the groups, while rituximab and IA were significantly more likely to have been used in the PNAb-positive group. PLEX aside, there was, however, no statistically significant difference in the clinician reported responses to these therapies between the 2 groups, although there was a trend to rituximab being more often judged effective in the PNAb-positive compared to PNAb-negative group.”  

We acknowledge in the discussion the limitations of these comparisons and the potential confounds.

  1. It is not clear if the outcome is defined on the basis of a neurological examination or if it is only subjective (referred from the patient). Due to the retrospective nature of the study, the small sample and the above mentioned limitations, I think that is really difficult to draw conclusions from these data. 

We have clarified that the assessment of response in this cohort is based on the treating clinicians’ overall subjective judgement (see L20, L109, L126, L220, L230/5) and further clarified we are comparing the physicians’ impression of response rather than making a genuine assessment of treatment efficacy (L254-256)

 “It should also be emphasised this is a comparison of the physicians’ overall impression of response, rather than an evaluation of the true efficacy or otherwise of these treatments.“

We agree that we are limited in our ability to assess outcomes in this cohort, as well as in the assessment of whether these are significantly different.  This is not the message we intend to communicate from these results, and we have further emphasised this. L409 to 416 of the discussion now states (L409-416):

“We emphasise that the evaluation of the PNAb cohort is limited by the retrospective and subjective nature of the patient assessment. In addition, the small number of cases precludes us from reaching any conclusions regarding the objective benefits of one treatment modality compared to the other in this setting. In addition, improvement in neurological symptoms following IA/PLEX may occur after a delay, which may not be reflected in the immediate judgement of the treating physician. Blinding, randomisation, standardised follow up, as well as a control group to judge the natural history of these heterogeneous diseases, are required for a definitive evaluation of apheresis treatment efficacy in these patient groups.”

However, we maintain that treating physicians are less likely to rate PLEx or IA as effective (when assessed in a global and subjective way) in PNAb positive compared to PNAb negative subjects and that this is important for clinical decision making (a response to PLEx or IA doesn’t necessarily mean the patient is paranodal antibody positive, and vice versa) and we have stated this directly (L416-418):

“However, it is notable that treating physicians were less likely to think that apheresis had been effective in PNAb-positive patients.”

Round 2

Reviewer 1 Report

The authors have revised the manuscript addressing all of my comments from the first review. I have no further comments or concerns.

Reviewer 2 Report

Dear authors, 

thank you for incorporating my suggestions. Many points have been now clarified. I don't have any other concerns about this paper.